# Endoplasmic Reticulum Stress Differently Modulates the Release of IL-6 and IL-8 Cytokines in Human Glial Cells

**DOI:** 10.3390/ijms25168687

**Published:** 2024-08-09

**Authors:** Paulina Sokołowska, Anna Wiktorowska-Owczarek, Jakub Tambor, Sebastian Gawlak-Socka, Edward Kowalczyk, Marta Jóźwiak-Bębenista

**Affiliations:** Department of Pharmacology and Toxicology, Medical University of Lodz, Żeligowskiego 7/9, 90-752 Lodz, Poland; anna.wiktorowska-owczarek@umed.lodz.pl (A.W.-O.); jakub.tambor@student.umed.lodz.pl (J.T.); sebastian.gawlak-socka@student.umed.lodz.pl (S.G.-S.); edward.kowalczyk@umed.lodz.pl (E.K.)

**Keywords:** astrocytes, microglia, tunicamycin, ER stress, proinflammatory cytokines

## Abstract

Endoplasmic reticulum (ER) stress is a significant player in the pathophysiology of various neurodegenerative and neuropsychiatric disorders. Despite the established link between ER stress and inflammatory pathways, there remains a need for deeper exploration of the specific cellular mechanisms underlying ER stress-mediated neuroinflammation. This study aimed to investigate how the severity of ER stress (triggered by different concentrations of tunicamycin) can impact the release of proinflammatory cytokines IL-6 and IL-8 from astrocytes and microglia, comparing the effects with those induced by well-known immunostimulants—tumor necrosis factor alpha (TNF-α) or lipopolysaccharide (LPS). Mild ER stress has a distinct effect on the cytokine release compared to more intense stress levels, i.e., diminished IL-6 production was accompanied by an increase in IL-8 level, which was significantly more pronounced in astrocytes than in microglia. On the contrary, prolonged or more severe ER stress induced inflammation in glial cells, leading to a time- and concentration-dependent buildup of proinflammatory IL-6, but unlike inflammatory agents, an ER stress inducer diminished IL-8 secretions by glial cells. The differences could hold importance in identifying ER stress markers as potential drug targets for the treatment of neurodegenerative diseases or mood disorders, yet this requires confirmation in more complex animal studies.

## 1. Introduction

The endoplasmic reticulum (ER) plays a pivotal role in regulating numerous biological functions, among others, the synthesis, folding, maturation, and transport of proteins. The disruption of ER function leads to the buildup of unfolded and misfolded proteins in the ER, a cellular state known as ER stress [1,2]. To restore ER homeostasis, the unfolded protein response (UPR) pathway is activated, initially resulting in adaptive programs that lead to the cessation of protein translation, degradation of misfolded proteins, enhanced protein folding, and synthesis of molecular chaperones crucial for proper protein folding. When the damage becomes irreversible, the UPR pathway leads to cell death [3]. These divergent outcomes depend on the duration and intensity of the stimuli, leading to conditions of mild, moderate, or severe ER stress [4,5]. During prolonged or severe ER stress, the objective of the UPR shifts from promoting cellular survival to triggering abnormal inflammatory signaling and apoptotic pathways [5,6,7,8]. In the central nervous system (CNS), the persistent activation of ER stress has been associated with various neurodegenerative diseases (e.g., Creutzfeldt–Jakob disease, Alzheimer’s disease, Parkinson’s disease, and Huntington’s disease) [9,10,11,12] and mood disorders, such as bipolar disorder [13] and major depressive disorder (MDD) [14]. In conditions such as stroke, ischemia, spinal cord injury, and amyotrophic lateral sclerosis, the prolonged activation of ER stress is considered a primary mechanism leading to neuronal disorders [9].

It is important to note that the activation of ER stress in neurons is a significant factor participating in their dysfunction. Stress reactions may be further affected by adjacent glial cells through glia–neuron crosstalk [11] Among glial cells, astrocytes represent the most abundant population, playing an essential supportive role in brain function. Astrocytes secrete factors crucial for neuronal survival, neurite outgrowth, neurotransmission, and metabolic regulation. However, in response to ER stress, astrocytes generate inflammatory mediators, diminish trophic support, and have the capability to transmit ER stress to neighboring cells. Reactive astrocytes also contribute to inflammatory responses by stimulating microglial activation [15]. Microglia, serving as resident macrophages in the CNS, actively monitor the brain’s microenvironment and play key roles in synaptic refinement, injury recovery, homeostasis regulation, phagocytic activity, and intercellular communication. In response to injuries, inflammatory stimuli, ER stress, or other insults, microglia are crucial mediators of neuroinflammation due to their ability to initiate or influence a wide range of cellular reactions. Changes in microglial function are associated with both brain development and aging, as well as various neurodegenerative conditions [16]. Dysregulated astrocytes and microglial cells promote events that lead to a neurotoxic microenvironment [17,18]. Understanding the complex interplay between astrocytes, microglia, and neuronal populations is crucial for unraveling the mechanisms underlying CNS disorders and developing targeted therapeutic interventions.

Neuroinflammation is involved in the onset and progression of neurodegenerative diseases, leading to neuronal loss. Additionally, peripheral inflammation exacerbates neuroinflammatory pathways by activating glial cells and neurons and increasing the blood–brain barrier (BBB) permeability [19]. The role of IL-6 in the pathogenesis of Alzheimer’s disease has been extensively studied and summarized in numerous meta-analyses. Interestingly, IL-6 appears to be associated with an increased risk of developing all-cause dementia, but this marker is not specific to Alzheimer’s disease [20,21,22]. However, polymorphisms in IL-6 –174 G/C, IL-6 –572 G/C, and IL-10 –1082 A/G may influence the predisposition to this condition [23]. Elevated serum levels of IL-6 have been observed in patients with Parkinson’s disease and are linked to non-motor symptoms and cognitive dysfunction [24]. In contrast, plasma IL-6 levels correlate with the progression of motor symptoms in Huntington’s disease and may serve as a biological marker for clinical applications [25].

The role of IL-8 in the development of neurodegenerative diseases is less understood compared to IL-6. The IL-8 gene polymorphism – 251T>A may contribute to the susceptibility to Alzheimer’s disease [26], and IL-8 levels are significantly higher in Parkinson’s disease patients [27]. However, IL-8 is a key marker in the progression of multiple sclerosis. Cerebrospinal fluid IL-8 concentrations are significantly higher in relapsing-remitting MS patients with the T variant of rs2227306 [28]. Additionally, serum IL-8 and IL-8 secretion from peripheral blood mononuclear cells (PBMCs) are significantly higher in multiple sclerosis patients. Thus, IL-8 may serve as a marker of monocyte activity in multiple sclerosis and could play a role in the recruitment of monocytes to the central nervous system [29].

While many studies have highlighted the direct link between ER stress and inflammatory pathways, deeper exploration into the specific cellular mechanisms through which ER stress exacerbates neuroinflammation is still necessary. Defining the nature of ER stress in different cell types in the brain, particularly with respect to induction or suppression of inflammatory signals, may contribute to filling the gap between experimental and clinical results. Targeting ER stress as a therapeutic approach for neurodegenerative diseases or psychiatric disorders requires markers for drug development or monitoring disease progression. Therefore, the aim of this study was to provide a comprehensive analysis of the effects of different forms of ER stress (mild, moderate, and severe) on the release of proinflammatory cytokines IL-6 and IL-8 from astrocytes and microglial cells. Tunicamycin was used as an inducer of ER stress due to its ability to hinder the formation of protein N-glycosidic linkages by blocking the transfer of N-acetylglucosamine 1-phosphate to dolichol monophosphate [30]. Furthermore, the effect of tunicamycin on cytokine release was compared to well-known immunostimulants—lipopolysaccharide (LPS) and tumor necrosis factor alpha (TNF-α)—to investigate differences in cytokine secretion under the influence of inflammatory factors and an ER stress inducer.

## 2. Results

### 2.1. Effect of Tunicamycin on Gene Expression Related to ER Stress in Glial Cells

Tunicamycin is a well-known inducer of ER stress; nevertheless, we confirmed its ability to stimulate the most representative genes within the classical UPR pathway in human microglial HMC3 cells (Table 1). We treated microglia for 24 h with tunicamycin at 0.5 µg/mL and compared the results with our previous findings on human astrocytes [20]. Similar to the previous studies, most ER stress-responsive genes displayed markedly increased expression levels, with notable upregulation of HSPA5 and DITT3 genes exhibiting fold changes of 10.14 and 12.22, respectively (Table 1).

### 2.2. Induction of Different Levels of ER Stress in Astrocytes and Microglia

To trigger various ER stress conditions in astrocyte and microglial cells, tunicamycin was administered at a wide range of concentrations for 24, 48, and 72 h. Cell viability was decreased in a time- and concentration-dependent manner (Figure 1). Based on our results, we propose that mild ER stress can be defined as non-lethal with no or minimal increase in cell viability, whereas moderate stress can be characterized by a statistically significant reduction in cell viability without cytotoxic effects, i.e., cell viability does not significantly decrease below the 70% threshold established by [32] for the MTT test as the boundary between the absence and potential occurrence of cytotoxicity. Severe ER stress leads to cytotoxicity with a decrease in viability below 70% compared to the untreated control. Mild ER stress in astrocytes was induced by tunicamycin at 0.001–0.01 µg/mL at all time points, i.e., 24, 48, and 72 h, leading to a significant increase in cell viability at the lowest tunicamycin concentration (Figure 1A–C). Microglia viability was not enhanced, but it was also not reduced across a wider range of tunicamycin concentrations (i.e., 0.001–0.5 µg/mL for 24 h, 0.001–0.1 µg/mL for 48 h, and 0.001–0.05 µg/mL for 72 h) (Figure 1D–F).

### 2.3. Impact of Different Levels of ER Stress on Cytokine Release in Astrocytes and Microglia

Astrocytes and microglia exhibited distinct reactions to tunicamycin regarding the release of IL-6. After 24 h, under mild ER stress conditions, the release of IL-6 from astrocytes was diminished compared to the untreated control (Figure 2A). When ER stress reached a moderate level (i.e., tunicamycin concentrations from 0.05 to 1 µg/mL), the IL-6 secretion from astrocytes underwent a statistically insignificant increase (Figure 2A). After 48 h, the IL-6 release from astrocytes was slightly higher compared to the untreated control and statistically significant in the range of tunicamycin concentrations from 0.05 to 1 µg/mL (Figure 2B). Prolonged ER stress (72 h) resulted in a similar profile of IL-6 release, but results were not statistically significant (Figure 2C). Unlike astrocytes, microglial cells exhibited a markedly higher IL-6 secretion in response to tunicamycin stimulation, reaching levels of approximately 800 pg/mL after 48 and 72 h of ER stress, as opposed to the approximately 40 pg/mL observed in astrocytes (Figure 2E,F vs. Figure 2B,C). Additionally, microglial cells responded with a statistically significant increase in IL-6 levels as early as 24 h (Figure 2D).

The IL-8 release in response to tunicamycin was comparable between both types of glial cells. Mild ER stress caused by low tunicamycin concentrations (i.e., 0.001–0.05 µg/mL) resulted in a significant rise in IL-8 level compared to the untreated control after 24 h in astrocytes and in a lesser IL-8 production in microglial cells (Figure 2A,D). This effect was not observed after longer 48 and 72 h tunicamycin stimulation (Figure 2B,C,E,F). Interestingly, severe ER stress significantly reduced IL-8 release in both cell types (Figure 2B,C,E,F).

### 2.4. Distinct Cytokine Secretion by Astrocytes and Microglia under the Influence of Inflammatory Agents

Studies on cytokine secretion upon the administration of positive controls (well-known immunostimulants), specifically TNF-α (0.05 ng/mL) for astrocytes and LPS (10 ng/mL; 800,000 EU/mL) for microglia, produced interesting results. IL-6 secretion showed a significant increase when stimulated by TNF-α in astrocytes and LPS in microglia compared to stimulation triggered by tunicamycin (Figure 3A,C). Conversely to the ER stress inducer, both inflammatory factors elicited a significant increase in IL-8 release from both types of glial cells (Figure 3B,D). Inflammatory agents caused a consistent increase in the levels of both cytokines at each time point, i.e., after 24, 48, and 72 h, whereas ER stress showed a variable course depending on its severity (mild, moderate, or severe) and the duration of exposure to the ER stress inducer (Table 2).

## 3. Discussion

The balance between ER stress and the folding capacity of the ER is crucial in managing the transition from an adaptive to a dysfunctional response. Erguler and co-workers proposed a combined mechanistic model encompassing the three signaling pathways of the UPR cascade (i.e., IRE1α, PERK, and ATF6) showing the UPR response in three distinct states of behavior: low, intermediate, and high activity states [33]. The states were linked to stress adaptation, tolerance, and the onset of apoptosis. The analysis demonstrated that the intermediate state may display fluctuations in translation inhibition and apoptotic signals; therefore, preconditioning that involves the application of specific ER stress inducers to promote an adaptive response may prevent the harmful effects of cell death [33,34]. These results gave evidence that ER stress is a dynamic process that can be regulated to mitigate its adverse effects or to promote adaptation processes. In the context of these studies, it is worth emphasizing that mild ER stress can have beneficial effects in the CNS. Mild ER stress induced by tunicamycin was protective in the Parkinson’s 6-OHDA mouse model, indicating that keeping the UPR response at a moderate level could help prevent the disease [35]. Similarly, Wang et al. showed that low doses of tunicamycin resulted in a mild ER-stress response without inducing cytotoxicity and tissue toxicity. Mild ER-stress preconditioning reduced microglia and astrocyte activation and neuronal death, as well as improved LPS-induced BBB impairment and cognitive ability dysfunction in rats [5,36].

For the above reasons, our goal was to determine concentrations of tunicamycin responsible for inducing a specific severity of ER stress. Analysis of the expression of genes belonging to three UPR signaling pathways confirmed the role of tunicamycin as an ER stress inducer in microglial cells. The same concentration of tunicamycin (i.e., 0.5 µg/mL) was used as in our previous studies to compare its effects in different cell types. We selected genes that are considered markers of ER stress, i.e., HSPA5 encoding GRP78 protein or DDIT3 translated into CHOP. The results showed upregulation of most genes selected in both glial cells with the highest expressions of marker genes, confirming the activation of the UPR pathway under the influence of the applied tunicamycin concentration (Table 1).

Cell viability measurement revealed a time- and concentration-dependent relationship with tunicamycin, allowing us to determine the degree of ER stress activation. The adopted classification into mild, moderate, and severe ER stress based on cell viability was in line with the work of Wang and colleagues, who also used cell viability or cell apoptosis measurements to identify the concentration corresponding to mild, non-cytotoxic ER stress and more severe ER stress resulting in cell death. Furthermore, the authors of the work showed variable expression of proteins of the UPR pathway depending on the intensity of ER stress, e.g., the hippocampal expression levels of ATF4 and CHOP were increased only at the highest dose of TM (30 μg), which induced cell death. At lower tunicamycin doses, i.e., 0.3 and 3 μg, no altered expressions of these proteins were observed [5]. Similarly, in cultured rat astrocytes, lower doses of tunicamycin (0.1 and 1 ng/mL), in contrast to the concentration of 10 ng/mL, had no effect on ATF4 and CHOP expression levels [36]. These results indicate that the analysis of the expression of the UPR pathway components may represent a way to identify the extent of ER stress severity in the CNS.

The possibility of developing new therapies supporting mild ER stress or preventing the harmful effects of chronic ER stress appears probable. The development of new therapies involves the search for specific markers that enable monitoring of therapeutic or toxic effects. One possibility is to measure levels of specific proteins involved in the UPR response to different severities of ER stress. However, due to a direct link between ER stress and inflammation that manifested in the ability of ER stress to regulate inflammatory pathways and resulted in cytokine production [8], we applied a different research approach and investigated whether the levels of IL-6 and IL-8 secreted by glial cells under ER stress can change depending on ER stress severity.

IL-6 is a 26 kDa proinflammatory cytokine produced by a variety of cells. It is an activator of acute phase responses, and the overproduction of IL-6 was seen in a variety of chronic autoimmune and inflammatory diseases [37]. IL-6 is also a major cytokine in the CNS, playing a role in regulating neuronal development, survival, and function, but it is also involved in neuroinflammation accompanying CNS diseases [38,39]. The crosstalk between ER stress and IL-6 production in the brain was widely described. For example, in primary murine astrocytes, the PERK-mediated arm of the UPR was the most linked to the induction of inflammation and IL-6 release through interaction with the Janus kinase 1 (JAK1)/signal transducer and activator of transcription 3 (STAT3) signaling pathway [40]. Additionally, free IL-6 could bind to its cell membrane receptor and further activate JAK1/STAT3, amplifying inflammation [41]. In microglia, both PERK/CHOP and IRE1a/XBP1s pathways were involved in modulating IL-6 secretion [42].

IL-8 is primarily known for its proinflammatory properties, as it acts as a chemoattractant for neutrophils and other immune cells, promoting inflammation and tissue damage in response to infection or injury [43]. The role of IL-8 in ER stress in the CNS has not been well understood so far, despite its involvement in neuroinflammation associated with schizophrenia [41], major depressive disorder [44], Alzheimer’s disease [45], or Parkinson’s disease [46]. IL-8 is synthesized and released by macrophages and brain microglia and astrocytes [47] and may serve either a pro- or anti-inflammatory role, mainly depending on the concentration [48]. IL-8 is produced early in the inflammatory response, possibly persisting for days or weeks, unlike most other inflammatory cytokines that are produced and cleared within a few hours. Thus, IL-8 might be specific for more chronic inflammatory changes in neurodegenerative and neuropsychological alterations in the brain [49]. Some links between ER stress marker genes and cytokines were described. For example, Krupkova et al. observed a positive correlation between the expression of the GRP78 and IL-6 and IL-8 genes, whereas no such correlation was noted for IL-1β or TNF-α in the human intervertebral disc [50].

As positive controls, we used TNF-α and LPS for astrocytes and microglia, respectively, due to their cell type-dependent response to different inflammatory inducers. According to the study of Ehrlich et al., fetal and adult human microglia exhibit the highest response to IL-8 secretion when stimulated by LPS, followed by IL-1β, and least to TNF-α; conversely, fetal astrocytes show the strongest secretion of IL-8 following IL-1β administration, followed by TNF-α, with no effect observed under LPS stimulation [47]. As expected, both inflammatory factors elicited a significant increase in IL-6 and IL-8 release from both types of glial cells (Figure 3).

The dose- and time-dependent analysis of the tunicamycin-induced release of IL-6 and IL-8 demonstrated interesting results. Mild ER stress (tunicamycin at concentrations of 0.001–0.05 µg/mL) reduced IL-6 release and significantly increased IL-8 production in human astrocytes after 24 h, but this effect was transient and was not observed with longer exposure times, i.e., 48 and 72 h (Figure 2 and Table 2). In contrast, at the lowest tunicamycin concentration (0.001 µg/mL), astrocyte viability increased at all time points, indicating that mild ER stress might have a long-term beneficial effect (Figure 1). In another study, mild ER stress diminished LPS-induced astrocytic inflammatory responses and overactivation, confirming its protective role in the CNS [35]. In microglial cells, IL-8 production induced by tunicamycin at concentrations of 0.001–0.05 µg/mL was only slightly increased after 24 h of exposure to mild ER stress without reaching statistical significance, and there was no change in IL-6 levels compared to the control. Given that ER stress is a dynamic process, further incubation of astrocytes and microglia with higher concentrations of tunicamycin led to a significant increase in IL-6 secretion and, interestingly, a decrease in IL-8 levels in both types of glial cells (Figure 2). This phenomenon was observed even at cytotoxic concentrations of an ER stress inducer, with cell viability below 70%, indicating that severe and prolonged ER stress triggers an inflammatory response in glial cells. Thus, the inflammatory response induced by tunicamycin at higher concentrations differs from that induced by LPS or TNF-α with regard to IL-8, but the significance of these results needs further research.

The limitation of this study is that it shows the effects of varying levels of ER stress on cytokine secretion only in cell cultures. Further research is needed in more complex animal models to check whether cytokine levels in the blood can also fluctuate under ER stress depending on its severity. This could help in understanding the potential significance of the findings presented in this study. For example, such confirmation at the organism level would enable the use of cytokine level measurements as predictive markers for monitoring ER stress severity and could aid in the development of new therapeutic strategies in research. Furthermore, maintaining a balanced level of cytokines appears to be crucial in the management of disorders induced or accompanied by inflammation. Numerous studies have substantiated the involvement of IL-6 and IL-8 in the pathogenesis of neurodegenerative diseases, establishing their utility as clinical markers. Therefore, therapeutic interventions aimed at restoring ER stress-disrupted equilibrium between IL-6 and IL-8 could hold promise in alleviating ER stress-related complications.

## 4. Materials and Methods

### 4.1. Reagents

Reagents for astrocyte culture, i.e., medium, Fetal Bovine Serum (FBS), growth supplement, penicillin/streptomycin solution, and poly-L-Lysine, were from ScienCell Research Laboratories (Carlsbad, CA, USA). Reagents for microglial cell culture—Eagle’s Minimum Essential Medium (EMEM), FBS, Dulbecco’s Phosphate Buffered Saline (D-PBS), and Trypsin-EDTA solution—were obtained from ATCC (Manassas, VA, USA). Tunicamycin, LPS (800,000 EU/mL), Trypsin-EDTA solution, and MTT (3-(4,5-dimethylthiazol-2-yl)-2,5-diphenyltetrazolium bromide) were from Sigma-Aldrich (Saint Louis, MO, USA). The RNeasy Mini Kit was obtained from Qiagen (Germantown, WI, USA), and other reagents for real-time PCR, i.e., Custom PrimePCR™ Real-Time PCR Plates, iScript™ cDNA Synthesis Kit, 2xSsoAdvanced Universal SYBR Green Supermix, Prime PCR RT Control, and Prime PCR Control Assay, were from Bio-Rad (Berkeley, CA, USA). Human IL-6 DuoSet ELISA and Human IL-8 DuoSet ELISA kits and Recombinant Human TNF-alpha Protein were from R&D Systems (Minneapolis, MN, USA).

### 4.2. Cell Culture

The astrocyte cell line from human cerebral cortex was purchased from ScienCell Research Laboratories (San Diego, CA, USA) and maintained according to the protocol recommended by the company. The microglial HMC3 cell line from human brain (Cat. No (CRL 3304) was obtained from ATCC (Manassas, VA, USA) and cultured according to the recommended protocol.

### 4.3. Analysis of Gene Expression Related to ER Stress

Analysis of gene expression was performed according to the method described in our previous work [20] to compare the results with those obtained for astrocyte cell culture. Briefly, microglial cells were seeded onto culture flasks at a density of 1–1.5 × 10^6^ cells/flask and cultured for 24 h. Following this, tunicamycin (0.5 µg/mL) was added to the culture, and cells were further incubated for 24 h. After the incubation period, cells were harvested, and the cell pellets were resuspended in RLT lysis buffer for RNA isolation and purification using the RNeasy Mini Kit from Qiagen. Reverse transcription was performed using the iScript™ cDNA Synthesis Kit in a CFX96 thermal cycler (Bio-Rad, Berkeley, CA, USA). The resulting cDNA and the 2xSsoAdvanced Universal SYBR Green Supermix reagent were added to appropriate wells on a 96-well Custom PrimePCR™ Real-Time PCR Plate according to the manufacturer’s protocol (Bio-Rad, Berkeley, CA, USA). The real-time PCR reaction was conducted in a CFX96 thermal cycler (Bio-Rad, Berkeley, CA, USA), and data analysis was performed using the 2^−ΔΔCq^ method. GAPDH and TBP were chosen as reference genes for normalization, and statistical significance was determined by one-way ANOVA with a significance level of *p* < 0.05. A fold change value above 1.5 was considered significant.

### 4.4. Cell Viability

The colorimetric MTT assay was used for the evaluation of cell viability under conditions of ER stress. Both types of glial cells were seeded onto 96-well plates at a final density of 7 × 10^3^ cells/well. Following 24 h of culture, the cells were treated with tunicamycin (0.001–5 μg/mL) for 24, 48, or 72 h. After the respective incubation periods, MTT solution was added to the cell culture for another four hours. Absorbance was measured at 570 nm using a BioTek EL ×800 microplate reader (BioTek, Winooski, VT, USA), with the value being directly proportional to the number of viable cells. Cell viability was calculated using the following formula: Viability [%] = (A/AC) × 100%, where A represents the absorbance of the investigated sample, and AC represents the absorbance of the control (untreated cells).

### 4.5. ELISA Tests

Astrocytes and microglia were seeded onto 12-well plates at a density of 1.5 × 10^5^ cells/well. After 24 h, the cells were exposed to tunicamycin (0.001–5 μg/mL) or LPS (10 ng/mL) or TNF-α (0.05 ng/mL) for 24, 48, or 72 h. Following the incubation periods, the supernatants of the cells were collected, and levels of IL-6 or IL-8 were measured using the Human IL-6 DuoSet ELISA or Human IL-8 DuoSet ELISA kit. Absorbance was measured at 450 nm using a BioTek EL ×800 microplate reader (BioTek, Winooski, VT, USA).

### 4.6. Data Analysis

Data are expressed as mean ± standard error of the mean (SEM). The results were tested by one-way ANOVA followed by post hoc Tukey’s multiple comparisons test or an unpaired *t*-test. For the calculations, GraphPad InStat version 9.3.0 (GraphPad, San Diego, CA, USA) was used.

## Figures and Tables

**Figure 1 ijms-25-08687-f001:**
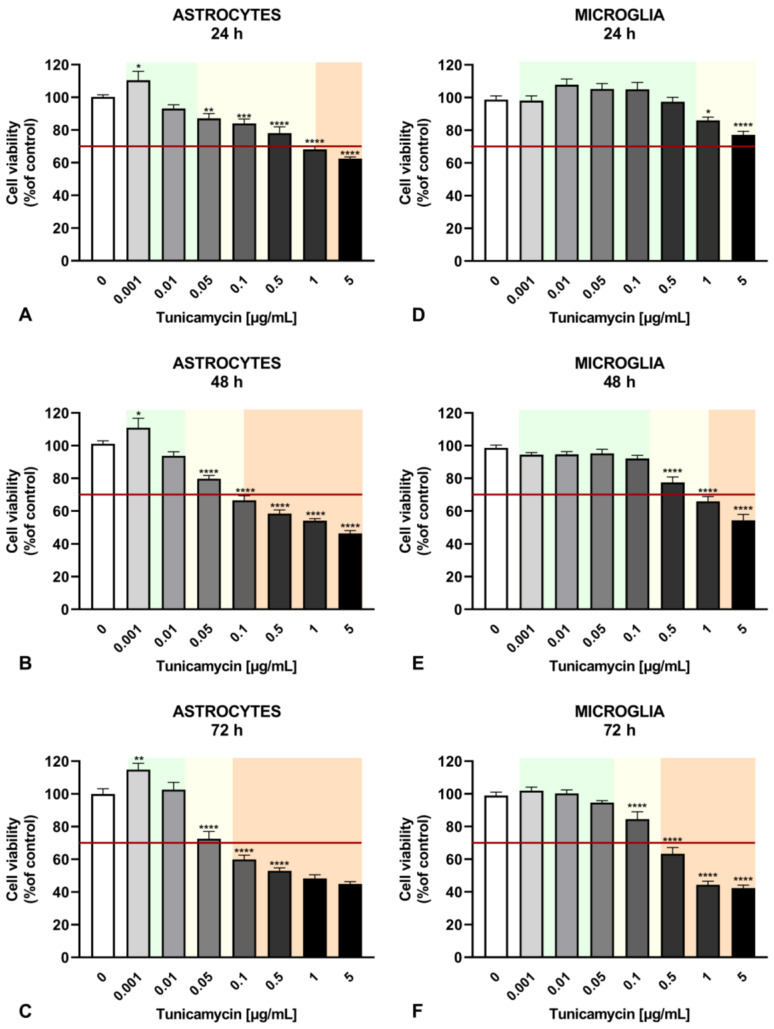
Effects of tunicamycin on the viability of astrocytes following 24 h (**A**), 48 h (**B**), and 72 h (**C**) of tunicamycin-induced ER stress, and on the viability of microglial cells after 24 h (**D**), 48 h (**E**), and 72 h (**F**) of tunicamycin stimulation. Green background—mild ER stress, yellow background—moderate ER stress, orange background—severe ER stress. Data are presented as the mean ± SEM and expressed as a percentage of untreated control cells. Statistical significance vs. control cells is indicated when appropriate; * *p* < 0.05; ** *p* < 0.01; *** *p* < 0.001; **** *p* < 0.0001.

**Figure 2 ijms-25-08687-f002:**
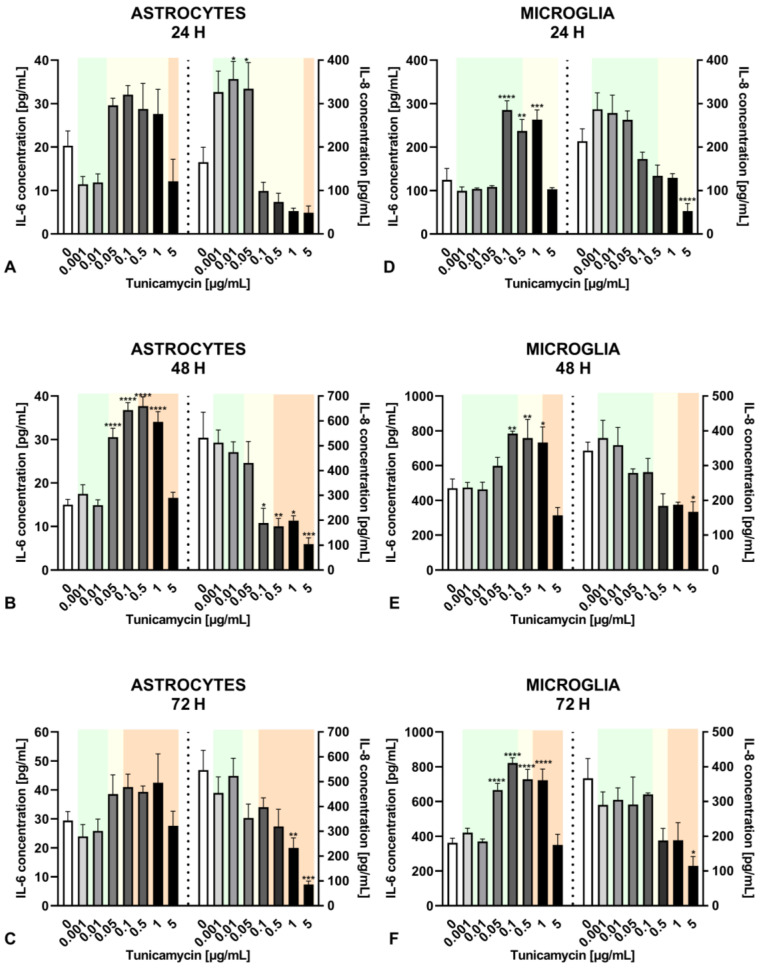
Effects of tunicamycin on IL-6 and IL-8 (**A**–**C**) release from astrocytes. Tunicamycin-induced IL-6 and IL-8 (**D**–**F**) release from microglial cells. Green background—mild ER stress, yellow background—moderate ER stress, orange background—severe ER stress. Data are presented as the mean ± SEM and expressed as a concentration of respective protein. Statistical significance vs. control cells is indicated when appropriate; * *p* < 0.05; ** *p* < 0.01; *** *p* < 0.001; **** *p* < 0.0001.

**Figure 3 ijms-25-08687-f003:**
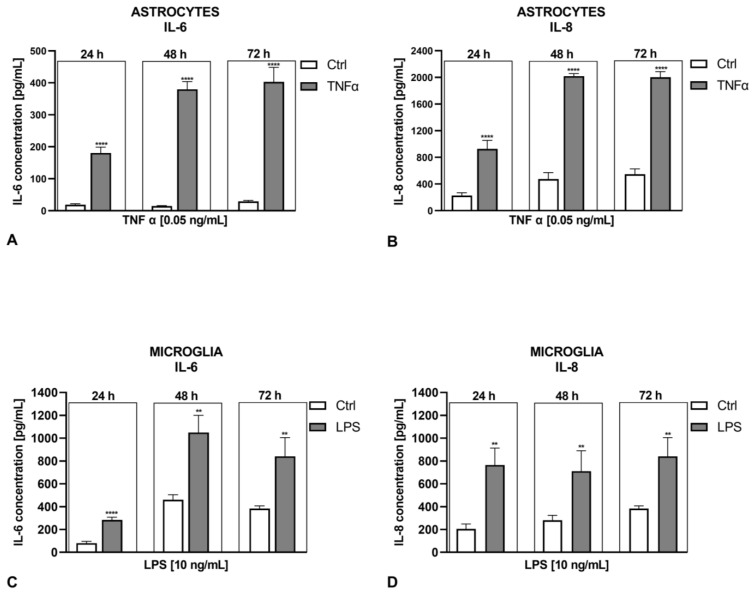
Effects of TNF-α on IL-6 release (**A**) or IL-8 secretion from astrocytes (**B**). Impact of LPS (10 ng/mL; 800,000 EU/mL) on IL-6 release (**C**) or IL-8 secretion (**D**) from microglial cells. Data are presented as the mean ± SEM and expressed as a concentration of respective protein. Statistical significance vs. control cells is indicated when appropriate; ** *p* < 0.01; **** *p* < 0.0001.

**Table 1 ijms-25-08687-t001:** Impact of tunicamycin on microglial cells in terms of the expression of ER stress-associated genes. Data are presented as the mean ± SEM and expressed as fold change vs. untreated control cells. * Results were shown for comparison.

Studied Gene	Alternative Name	Encoded Protein	MICROGLIA	ASTROCYTES [31] *
			Fold Change	Fold Change
			Tunicamycin [0.5 µg/mL]	Tunicamycin [0.5 µg/mL]
*ATF4*	*CREB-2*	Activating transcription factor 4	2.33 ± 0.14	2.87 ± 0.28
*ATF6*	*-*	Activating transcription factor 6	2.00 ± 0.52	2.50 ± 0.31
*CREB3L1*	*Oasis*	CAMP responsive element binding protein 3 like 1	1.17 ± 0.37	1.49 ± 0.13
*DDIT3*	*CHOP*	DNA damage-inducible transcript 3/C/EBP-homologous protein	12.22 ± 2.29	13.75 ± 1.08
*EDEM1*	*EDEM*	ER degradation enhancing alpha-mannosidase-like protein 1	1.83 ± 0.34	3.85 ± 0.35
*ERN1*	*IRE1*	Endoplasmic reticulum to nucleus signaling 1/Inositol-requiring enzyme 1	1.41 ± 0.45	3.61 ± 0.37
*HSPA5*	*GRP78*	Heat shock protein family A (Hsp70) member 5	10.14 ± 2.04	19.36 ± 1.26

**Table 2 ijms-25-08687-t002:** A comparison between the effect of ER stress severity and inflammatory agents (TNF-α and LPS) on IL-6 and IL-8 release by astrocytes and microglia. 
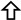
—increase; 
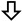
—decrease; 
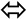
—no effect vs. untreated control.

ER Stress	MILD	MODERATE	SEVERE	TNF-α/LPS
ASTROCYTES
Hours	24	48	72	24	48	72	24	48	72	24	48	72
IL-6	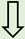	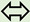	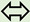	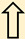	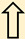	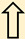	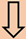	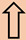 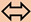	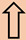 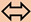	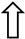	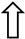	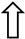
IL-8	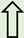	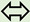	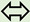	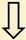	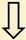	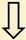	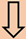	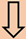	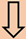	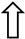	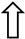	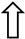
MICROGLIA
Hours	24	48	72	24	48	72	24	48	72	24	48	72
IL-6	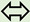 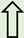	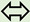 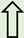	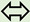 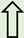	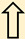 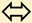	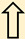	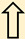	-	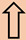 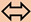	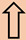 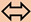	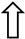	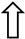	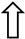
IL-8	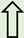 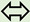	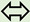	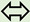	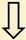	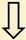	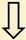	-	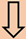	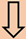	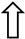	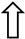	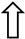

## Data Availability

The dataset is available on request from the authors.

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
