# Peer review of "Endoplasmic Reticulum Stress Differently Modulates the Release of IL-6 and IL-8 Cytokines in Human Glial Cells"

_ijms, 2024, doi:10.3390/ijms25168687_

Round 1
Reviewer 1 Report
Comments and Suggestions for Authors
In this study the authors investigated the differential impact of ER stress on the release of IL8 and IL6, two proinflammatory cytokines, in response to the immunostimulants - tumor necrosis factor alpha (TNF-α) or lipopolysaccharide (LPS). They found that mild ER stress diminished IL-6 production was accompanied by an increase in IL-8 levels. However, severe ER stress diminished IL-8 secretions by glial cells. While this is an interesting study, there some issues that need to be addressed:
1.In the abstract, it is not clear how ER stress was induced.
2. The study compared the levels of IL6 and IL8 in cultured microglia and astrocytes separately, it would be important to compare the levels of these inflammatory cytokines between astrocyte and microglia at each time point, for example in figure 2.
3. What is the reason for using LPS for microglia and TNFα for astrocytes as the immunostimulators? Why did they not try using LPS for both astrocytes and microglia?
4. In the aim of the study, the authors mentioned they compared the levels of IL8 and IL6 after different levels of ER stress, but they did not explain how they classified ER stress as mild, moderate and sever. Importantly, there is no plot showing the comparison between different levels of ER stress, TNFα and LPS.
Author Response
At the very beginning we would like to thank the Reviewers for their great commitment and accurate comments concerning our manuscript. We are convinced that all suggestions improve final quality of our paper. Below we present responses to all comments in order as in original review. All changes in revised version of the manuscript are highlighted in red.
Reviewer #1:
In this study the authors investigated the differential impact of ER stress on the release of IL8 and IL6, two proinflammatory cytokines, in response to the immunostimulants - tumor necrosis factor alpha (TNF-α) or lipopolysaccharide (LPS). They found that mild ER stress diminished IL-6 production was accompanied by an increase in IL-8 levels. However, severe ER stress diminished IL-8 secretions by glial cells. While this is an interesting study, there some issues that need to be addressed:
1.In the abstract, it is not clear how ER stress was induced.
Ad. 1 Information about tunicamycin was included in the abstract.
- The study compared the levels of IL6 and IL8 in cultured microglia and astrocytes separately, it would be important to compare the levels of these inflammatory cytokines between astrocyte and microglia at each time point, for example in figure 2.
Ad.2 Figure 2 was changed according to Reviewer’s suggestions. We combined both cytokines on a single graph for each time point using two Y-axes and different scales to clearly visualize the changes of cytokines levels in relation to tunicamycin concentration. Adding several additional plots to a single graph to include both types of glial cells made it unreadable. Therefore, the graphs showing cytokine levels for astrocytes and microglia were arranged side by side in a single panel to facilitate comparison. Additionally, addressing the reviewer's comment in point 4, we added a background to each graph representing different degrees of ER stress severity to facilitate tracking changes in cytokine levels depending on the ER stress. The background was selected based on cell viability data (Figure 1). Background explanations are provided in the figure legend.
- What is the reason for using LPS for microglia and TNFα for astrocytes as the immunostimulators? Why did they not try using LPS for both astrocytes and microglia?
Ad.3 In our preliminary studies, we tried to stimulate astrocytes with LPS, but the response was surprisingly low. This phenomenon may be due to the fact that human fetal astrocytes do not respond to LPS in the same manner as cells obtained e.g. from animals. The varied responses of human glial cells to immunostimulators were described in the literature and presented in the discussion part (please, see lines: 269-274).
- In the aim of the study, the authors mentioned they compared the levels of IL8 and IL6 after different levels of ER stress, but they did not explain how they classified ER stress as mild, moderate and sever. Importantly, there is no plot showing the comparison between different levels of ER stress, TNFα and LPS.
Ad.4 We classified ER stress as mild, moderate and severe on the basis of cell viability measured by MTT test. A more precise description of this classification was added in the Results section (please, see lines: 120-123). Mild ER stress was classified in the literature on the basis of the effect of low concentrations of tunicamycin which resulted in no effects on the cell viability [e.g. doi: 10.3389/fncel.2018.00222], whereas severe ER stress leads to cell death [e.g. doi: 10.3389/fncel.2018.00222]. For the purposes of this study, we introduced the concept of moderate ER stress as an intermediate process with boundaries that are difficult to define due to the dynamic nature of the ER stress. The criteria are based on the assessment of cell viability on a scale: no changes in cell viability (or a slight increase) - mild ER stress, statistically significant changes in cell viability (compared to untreated control) but not exceeding the 70% threshold established by PN-EN ISO 10993-5 for the MTT test as the boundary between absence and potential occurrence of cytotoxicity – moderate ER stress, and changes in cell viability significantly below 70% threshold resulting in potential cytotoxicity – severe ER stress. For better visualization of the relation between tunicamycin-induced reduction in cell viability and the severity of ER stress, we added colored backgrounds corresponding to the adopted criteria in Figure 1 and 2.
Preparing a plot showing the comparison between different levels of ER stress, TNFα and LPS proved too complex, therefore we proposed a table 2. We hope that this solution is acceptable.

Reviewer 2 Report
Comments and Suggestions for Authors
The authors have submitted a research article regarding an impact of various concentrations of tunicamycin (i.e. various levels of “ER stress”) on release of interleukins 6 and 8 from cell lines of astrocytes and microglia, as compared with NF-alpha and LPS instead of tunicamycin. The results of the present study indicate a possible different response of IL6/8 releases from the cell types to tunicamycin, illustrating a hypothesis suggesting the useful and effective predictor for neurodegenerative disorders associated with degrees of ER stress (i.e. concentrations of tunicamycin) in clinic. The authors discussed the beneficial availability of the levels of IL 6/8 releases in cell culture as an indicator of neurodegenerative disorders, resulting in the expected pharmacologic properties which ameliorate the states of the neurodegenerative disorders, with perspectives. This issue is of interest, and impact of their results may be strong. My overall concern with the article describing the current available data regarding beneficial availability of the evaluation of neurodegenerative disorders in terms of ER stress, offer something substantial that helps advance our understanding of effective further medicinal management available in clinic.
To strengthen authors’ perspectives, the authors are strongly recommended to describe clearly previous findings regarding how the levels of IL 6 or 8 are important in promoting b neurodegenerative disorders. This is precisely the scientific knowledge that readers most want to know about.
Author Response
At the very beginning we would like to thank the Reviewers for their great commitment and accurate comments concerning our manuscript. We are convinced that all suggestions improve final quality of our paper. Below we present responses to all comments in order as in original review. All changes in revised version of the manuscript are highlighted in red.
The authors have submitted a research article regarding an impact of various concentrations of tunicamycin (i.e. various levels of “ER stress”) on release of interleukins 6 and 8 from cell lines of astrocytes and microglia, as compared with NF-alpha and LPS instead of tunicamycin. The results of the present study indicate a possible different response of IL6/8 releases from the cell types to tunicamycin, illustrating a hypothesis suggesting the useful and effective predictor for neurodegenerative disorders associated with degrees of ER stress (i.e. concentrations of tunicamycin) in clinic. The authors discussed the beneficial availability of the levels of IL 6/8 releases in cell culture as an indicator of neurodegenerative disorders, resulting in the expected pharmacologic properties which ameliorate the states of the neurodegenerative disorders, with perspectives. This issue is of interest, and impact of their results may be strong. My overall concern with the article describing the current available data regarding beneficial availability of the evaluation of neurodegenerative disorders in terms of ER stress, offer something substantial that helps advance our understanding of effective further medicinal management available in clinic.
To strengthen authors’ perspectives, the authors are strongly recommended to describe clearly previous findings regarding how the levels of IL 6 or 8 are important in promoting b neurodegenerative disorders. This is precisely the scientific knowledge that readers most want to know about.
We would like to thank the Reviewer for the accurate comment. We added additional paragraph in the introduction (please, see lines: 65-85) and modified slightly conclusions (please, see lines:305-307).

Round 2
Reviewer 1 Report
Comments and Suggestions for Authors
Authors responded to the comments in a satisfactory manner.
Author Response
Reviewer 1
Authors responded to the comments in a satisfactory manner.
Response: We thank the Reviewers for their time and their precise feedback on our manuscript, which significantly helped improve the quality of our work.
Reviewer 2 Report
Comments and Suggestions for Authors
The authors have addressed properly all the issues raised by reviewers including me. I have no more comments, and now recommend that this manuscript is acceptable for publication in the journal IJMS.
Author Response
Reviewer 2
The authors have addressed properly all the issues raised by reviewers including me. I have no more comments, and now recommend that this manuscript is acceptable for publication in the journal IJMS.
Response:
We thank the Reviewers for their time and their precise feedback on our manuscript, which significantly helped improve the quality of our manuscript.